# Systemic Biomarkers and Unique Pathways in Different Phenotypes of Heart Failure with Preserved Ejection Fraction

**DOI:** 10.3390/biom12101419

**Published:** 2022-10-04

**Authors:** Hao Chen, Milorad Tesic, Valentina N. Nikolic, Milan Pavlovic, Rada M. Vucic, Ana Spasic, Hristina Jovanovic, Ivana Jovanovic, Stephanie E. L. Town, Matthew P. Padula, Lana McClements

**Affiliations:** 1School of Life Sciences & Institute for Biomedical Materials and Devices, Faculty of Science, University of Technology Sydney, Ultimo, NSW 2007, Australia; 2Clinic for Cardiology, University Clinical Center of Serbia, 11000 Belgrade, Serbia; 3Faculty of Medicine, University of Belgrade, 11000 Belgrade, Serbia; 4Faculty of Medicine, Department of Pharmacology and Toxicology, University of Nis, 18000 Nis, Serbia; 5Faculty of Medicine, Department of Internal Medicine—Cardiology, University of Nis, 18000 Nis, Serbia; 6Faculty of Medical Sciences, Department of Internal Medicine, University of Kragujevac, 34000 Kragujevac, Serbia; 7Department of Cardiology, Clinical Centre of Kragujevac, 34000 Kragujevac, Serbia

**Keywords:** heart failure, biomarkers, heart failure with preserved ejection fraction, HFpEF, HCM, hypertrophic cardiomyopathy, proteomics, plasma

## Abstract

Heart failure with preserved ejection fraction (HFpEF) accounts for around 50% of all heart failure cases. It is a heterogeneous condition with poorly understood pathogenesis. Here, we aimed to identify unique pathogenic mechanisms in acute and chronic HFpEF and hypertrophic cardiomyopathy (HCM). We performed unbiased, comprehensive proteomic analyses of plasma samples from gender- and BMI-matched patients with acute HFpEF (*n* = 8), chronic HFpEF (*n* = 9) and HCM (*n* = 14) using liquid chromatography–mass spectrometry. Distinct molecular signatures were observed in different HFpEF forms. Clusters of biomarkers differentially abundant between HFpEF forms were predominantly associated with microvascular inflammation. New candidate protein markers were also identified, including leucine-rich alpha-2-glycoprotein 1 (LRG1), serum amyloid A1 (SAA1) and inter-alpha-trypsin inhibitor heavy chain 3 (ITIH3). Our study is the first to apply systematic, quantitative proteomic screening of plasma samples from patients with different subtypes of HFpEF and identify candidate biomarkers for improved management of acute and chronic HFpEF and HCM.

## 1. Introduction

Approximately 50% of patients with heart failure (HF) present with a preserved left ventricular ejection fraction (LVEF), a condition referred to as heart failure with preserved ejection fraction (HFpEF) [1]. HF with reduced ejection fraction (HFrEF) accounts for the rest of the 50% of HF cases with better pharmacological options than HFpEF. As a multifactorial disease, HFpEF is associated with diastolic dysfunction, which can also develop with healthy ageing in the absence of cardiovascular diseases [2]. To date, no drugs have shown effectiveness in the management of patients with chronic HFpEF and prevention of acute HFpEF, which is likely due to the poorly understood pathogenesis of HFpEF [3].

Hypertrophic cardiomyopathy (HCM) is often a key precursor of HF. The progression of HCM into HFpEF was described in a large population-based observational study [4], in which patients with HCM and elevated levels of protein biomarkers of haemodynamic stress or myocardial injury had a substantially higher risk of developing HF. However, other than cardiac-specific conditions such as HCM, HFpEF is a heterogenous systemic syndrome with comorbidity-related pathophysiology. There is growing recognition of further HFpEF classification according to phenotypic heterogeneity [5]. Nevertheless, there is a lack of studies to comprehensively unravel the molecular differences between the different types of HFpEF, leading to unspecific diagnostic biomarkers and poor efficacy of pharmacotherapies in the management of HFpEF.

Current ‘omics studies investigating the molecular mechanisms of HF are mainly focused on HFrEF [6], while only two studies reported the proteome profiling of HFpEF [7,8]. One of these studies utilised cryopreserved human hearts [7], which are difficult to access and obtain in clinical practice, and do not necessarily reflect real-time pathophysiology. The other study is a pilot observational proteomic study without quantification of differentially abundant proteins and bioinformatic analyses [8]. Hence, a better understanding of the systemic proteome profiles of different forms of HFpEF is warranted.

Given the ease of access to plasma samples, the aim of this study was to identify perturbed proteins that could be translated into specific biomarkers for different forms of HFpEF including chronic and acute cases. We also identified associated pathways providing insight into the pathogenesis of different forms of HFpEF including HCF, acute and stable HFpEF patients.

## 2. Material and Methods

### 2.1. Collection of Plasma

A total of 31 gender- and BMI-matched patients diagnosed with HFpEF in accordance with the latest guidelines for HF [9] were consecutively enrolled and transthoracic echocardiography was performed for each patient. The blood samples were collected at one point in time and centrifuged at 3000× *g* for 10 min to obtain plasma. All participants provided written informed consent prior to inclusion. Ethical approvals were obtained from all hospitals and institutions involved in this study. The study was conducted in accordance with the Declaration of Helsinki.

The patients were stratified into acute HFpEF (*n* = 8), chronic HFpEF (*n* = 9) and HCM (*n* = 14) (Figure 1). Briefly, the HCM group was defined as a homogenous cardiac-specific phenotype of LV hypertrophy with varying degrees of diastolic dysfunction that might progress into HFpEF. The acute HFpEF group included patients who were admitted to the hospitals due to exacerbation and decompensation of HFpEF, whereas the chronic HFpEF group included patients with stable symptoms of HFpEF that present with common HFpEF-associated co-morbidities including diabetes mellitus and/or hypertension.

### 2.2. Proteomics

Plasma sample preparation was adjusted from the previously described methods using STop-And-Go-Extraction tips (STAGE Tips) [10,11], where peptides were prepared for LC-MS/MS analysis. The mass spectrometry was performed in accordance with the protocols described previously [12]. The MS/MS data files were searched using PEAKS Studio X+ (Bioinformatics Solutions, Waterloo, Canada). The results of the search were then filtered to include peptides with a −log_10_*p* score that was determined by the false discovery rate (FDR) < 1%, in which decoy database search matches were <1% of the total matches.

### 2.3. Statistics

All statistical analyses were performed in R (4.0.2). All proteomic data were total ion count (TIC)-normalised, followed by log_2_-transformed prior to entering pipelines in R. The packages or functions utilised in this study included (i) limma package [13] for differential abundance analysis of acute HFpEF vs. HCM, chronic HFpEF vs. HCM and acute HFpEF vs. chronic HFpEF groups, while adjusting for age, gender and diabetes, (ii) geneSetTest function in limma package for pathway analyses annotated by Reactome database [14], and (iii) igraph package for network analyses of protein–protein interactions (PPIs) annotated by pairwise Pearson correlation, PPI scores in the STRING database [15] and protein functions in DAVID database [16,17]. Relative quantification of proteins or pathways between subtypes was expressed as fold change (FC). The Benjamini-Hochberg method was used to calculate the adjusted *p* values, also known as FDR, for all aforementioned analyses. Differentially abundant proteins were defined as FDR < 0.01. The power of the study was based on at least five samples per group, which is needed to provide sufficient statistical power to detect relative changes in abundances > 1.5-fold [18].

### 2.4. Data Availability

The proteomics data have been submitted to ProteomeXchange Consortium with the identifier PXD024012. All significant data and detailed methods supporting the findings are available in Appendix A. Alternatively, we provided a publicly available, interactive online application of supporting data for easy navigation (https://hao-chen-uts-99171821.shinyapps.io/HFpEF-Proteomics/).

## 3. Results

### 3.1. Patient Characteristics

The gender and BMI distributions across the three subtypes were similar (Table 1). An older age range was observed in the acute HFpEF group compared to the HCM group (*p* = 0.002) in keeping with the frequent onset of adverse outcomes in patients with HFpEF at an older age. In accordance with New York Heart Association (NYHA) class, patients in acute HFpEF group had more severe symptoms than those in chronic HFpEF and HCM groups. In line with this observation, the levels of N-terminal pro-B-type natriuretic peptide (NT-proBNP) were higher in the acute HFpEF group compared to HCM group (*p* = 0.046) and there was a trend towards higher NT-proBNP in acute vs chronic HFpEF (*p* = 0.06).

### 3.2. Protein Markers

Quantitative label-free proteomic analysis of non-depleted plasma samples was conducted by measuring the relative abundance of tryptic peptides using data-dependent acquisition (DDA) mass spectrometry. The generated outputs contained 273 proteins detected among all 31 samples with 93% data completion. All significantly different proteins between the three groups (acute HFpEF vs. HCM, chronic HFpEF vs. HCM and acute HFpEF vs. chronic HFpEF) are presented in Appendix A.

Initially, the clustering of groups was assessed by observing the principal component analysis (PCA) plot (Figure 1a), revealing a good separation of different types of HFpEF. The heatmap (Figure 1b) is consistent with the PCA plot, where hierarchical clustering displayed a clear separation between three subtypes. Collective, distinct molecular pathophysiologies between the three different forms of HFpEF were observed.

The differential abundance analysis was performed by three separate comparisons, including: (i) acute HFpEF vs. HCM, (ii) chronic HFpEF vs. HCM and (iii) acute HFpEF vs. chronic HFpEF, while adjusting for age, gender and diabetes, the major confounders of cardiovascular diseases. In total, there were 46, 31 and 24 differentially abundant proteins between acute HFpEF vs. HCM (Appendix A), chronic HFpEF vs. HCM (Appendix A) and acute HFpEF vs. chronic HFpEF (Appendix A), respectively (Figure 1c–e and Figure 2a).

Generally, the proteomes in acute HFpEF vs. HCM were correlated with inflammation (Figure 2a); the differentially abundant proteins in chronic HFpEF vs. HCM were correlated with immune system perturbations (Figure 2a); and the difference in the proteome profile between acute and chronic HFpEF was mainly related to immune and haemostatic proteins (Figure 2a). The differential abundance analysis revealed an approximately 2-fold decrease in apolipoprotein A-I (APOA1) in both acute HFpEF vs. HCM (FDR = 1.79 × 10^−5^) and chronic HFpEF vs. HCM (FDR = 3.22 × 10^−6^). Leucine-rich alpha-2-glycoprotein 1 (LRG1), a regulator of angiogenesis [19], was elevated in all three comparisons, including acute HFpEF vs. HCM (FC = 2.32, FDR = 1.65 × 10^−6^), chronic HFpEF vs. HCM (FC = 1.60, FDR = 1.29 × 10^−3^) and acute vs. chronic HFpEF (FC = 1.45, FDR = 6.21 × 10^−3^), showing a gradual increase from HCM to chronic HFpEF to acute HFpEF (1:1.6:2.3). Apart from LRG1, immunoglobulin kappa variable 1–12 (IGKV1–12) is also shared across all three comparisons, and it is the most significant differentially abundant protein in chronic HFpEF vs. HCM (FC = 0.19, FDR = 6.42 × 10^−11^) and acute HFpEF vs. chronic HFpEF (FC = 3.05, FDR = 2.76 × 10^−7^).

The proteins with the most noticeable perturbations were highlighted in a three-dimensional plot (Figure 2b). C-reactive protein (CRP), an established biomarker of acute HFpEF, was discovered as one of the differentially abundant proteins with a significant fold change (log_2_FC = 8.96, FDR = 3.36 × 10^−4^) in acute HFpEF vs. HCM. Although von Willebrand factor (vWF), a well-known HFpEF biomarker linked to endothelial dysfunction, did not present with a substantial perturbation, it was a unique protein differentiated between acute HFpEF and chronic HFpEF (log_2_FC = 2.19, FDR = 3.54 × 10^−3^). Apart from the previously well-validated biomarkers for HFpEF including CRP and vWF, we also identified other differentially abundant proteins in HFpEF (Figure 2a,b). Similar to other studies which identified serum amyloid A1 (SAA1) as differentially abundant protein in human HF samples [8,20,21], we also demonstrated that SAA1 was perturbed in chronic HFpEF compared to HCM with a substantial fold change (log_2_FC = 7.82, FDR = 7.47 × 10^−3^).

### 3.3. Pathway Analyses

Pathways altered across the three types of HFpEF were determined using the Reactome database [14] and were assessed via a triple Venn diagram (Figure 3a). The Venn diagram revealed a series of altered haemostasis and protein metabolism pathways, all of which form a core set of regulated pathways in the pathogenesis of HFpEF.

A total of 27 pathways were significant with four pathways shared between all three comparisons (Figure 3a,b; Appendix A). We identified the perturbations of differentially abundant proteins in the biological pathway regulating insulin-like growth factor (IGF) transport and uptake through insulin-like growth factor binding protein (IGFBP) (Appendix A), which plays a significant role in all types of HFpEF, particularly in acute HFpEF vs. chronic HFpEF (FDR = 3.43 × 10^−3^).

Platelet degranulation in response to elevated platelet cytosolic Ca^2+^ was the most significant pathway altered between acute and chronic HFpEF (Figure 3c; FDR = 2.68 × 10^−5^), and between acute HFpEF and HCM (FDR = 2.68 × 10^−10^). Galectin-3 binding protein (Gal3bp) and vWF, proteins correlated with well-known biomarkers of HFpEF, are both elevated in the platelet degranulation pathway. Related to this pathway, in the comparison between acute HFpEF and either chronic HFpEF or HCM, complement factor D (CFD) and inter-alpha-trypsin inhibitor heavy chain 3 (ITIH3) were elevated in alpha and dense granules, respectively (Figure 2). A reduction in the abundance of apolipoprotein A-I (APOA1) and/or plasminogen (PLG) were also observed in acute HFpEF, compared to other forms of HFpEF.

The complement cascade pathway was among the most significantly perturbed pathways between chronic HFpEF and HCM (FDR = 2.54 × 10^−3^), likely reflective of the heterogeneity of HFpEF. Platelet regulation pathways related to fibrin clot formation were the most relevant pathways differentiating acute from chronic HFpEF (Figure 4, FDR = 2.68 × 10^−5^), highlighting the likely impaired haemostasis associated with acute exacerbation of HFpEF (Appendix A).

### 3.4. Network Analyses

Pairwise correlation network analysis was next performed to investigate PPIs between different HFpEF subtypes. Figure 4a–c showed networks highlighting the most correlated nodes (Pearson correlation coefficient [*r*] > 0.7), where the colour and the length of the edge are proportional to Pearson *r*. Overall, our data in terms of PPIs were consistent with those reported by broad evidence in the STRING database (Figure 4a–c), as indicated by the majority of thick, opaque edges. To aid the readability of networks, the size of each node was programmed proportional to the corresponding eigencentrality (the influence in the entire network; Appendix A).

As shown in the networks, most PPIs were positively correlated. Proteins associated with complement/coagulation/protease and extracellular matrix (ECM) were the key proteins across all groups. ECM proteins are slightly more influential in acute HFpEF vs. HCM and chronic HFpEF vs. HCM (Figure 4a,b). However, immune proteins were more notable in acute HFpEF vs. chronic HFpEF comparison (Figure 4c). Despite more proteins being present between chronic HFpEF vs. HCM than acute HFpEF vs. HCM, PPIs were distinctively more pronounced in acute HFpEF vs. HCM, indicating a more complex interplay of proteins between these two types of HFpEF.

## 4. Discussion

Using plasma samples from gender- and BMI-matched patients with different forms of HFpEF including HCM (cardiac-specific cause of HFpEF), chronic HFpEF (heterogeneous group) and acute HFpEF (hospitalised), we have identified a core set of plasma biomarkers and signalling pathways.

Aberrant Ca^2+^ homeostasis in cardiomyocytes is a hallmark intramyocardial pathogenic mechanism of HF. Nevertheless, therapies targeting this aberrant intramyocardial biological process (e.g., beta-blockers) are beneficial for HFrEF, while ineffective for HFpEF [22], suggesting a potential extramyocardial origin of HFpEF. This is strongly supported by an in vivo study where cardiac HFpEF-like features were transferrable through transfusing blood between mice [23]. Therefore, studying HFpEF pathogenesis using blood-derived samples could be equally informative as myocardial tissues, whilst being less invasive, easily accessible and more reflective of real-time changes in HF progression. In this study, we have identified novel pathways related to HFpEF subtypes, associated with perturbations in a series of complement/coagulation/protease, immune system and/or ECM proteins, highlighting the importance of haemostasis-associated mechanisms, notably platelet degranulation likely by systematically elevated intraplatelet Ca^2+^. This finding suggests potential novel pathogenesis extrinsic to the heart in HFpEF.

The effects on the myocardium as a result of systemic inflammation in HFpEF likely originates from systemic microvascular endothelial dysfunction. Our study demonstrated the importance of the platelet degranulation pathway in response to elevated intraplatelet Ca^2+^ induced by systemic microvascular impairment in HFpEF. Platelet activation is mediated by secreted molecules from alpha and dense granules. Within this pathway, we showed elevated abundance of a series of ECM proteins (e.g., vWF, Gal3bp and ITIH3) and decreased abundance of APOA1, F5 and PLG. The link between inflammation and endothelial dysfunction in cardiovascular disease is well-established; vWF is key in this process where it is secreted into the circulation from endothelial cells in response to systemic inflammation. Furthermore, vWF possesses intrinsic proinflammatory properties capable of attracting leukocytes to the inflammatory site within the endothelial barrier [24], therefore exacerbating the inflammatory state. Other proteins increased in abundance in this pathway are also associated with inflammation. Another important protein in this pathway is fibrinogen beta chain (FGB) which, together with alpha and gamma chains, forms fibrinogen, a key component of insoluble fibrin matrix in the clotting cascade [25]. Elevated abundance of FGB in acute or chronic HFpEF compared to HCM likely highlights thromboembolic events contributing to HFpEF hospitalisation and heterogeneity of HFpEF.

We have also shown a reduction in abundance in a number of key proteins including APOA1, the main component of high-density lipoprotein (HDL), in both acute and chronic HFpEF compared to HCM. External to the pathway of platelet activation, other apolipoprotein components of HDL (e.g., APOA4, APOD) were also reduced in abundance in acute or chronic HFpEF compared to HCM. Interestingly, serpin family A member 1 (SERPINA1) or alpha-1-antitrypsin was decreased in chronic HFpEF vs. HCM, and increased in acute HFpEF vs. HCM or acute vs. chronic HFpEF, suggesting that SERPINA1 levels may be substantially low in chronic HFpEF. Given that the SERPINA1-HDL complex has a protective effect against inflammation [26], a decrease in both SERPINA1 and HDL components in chronic HFpEF further exacerbates the deleterious effects originating from systemic inflammation.

All the perturbed proteins in the platelet degranulation pathway together highlight potential underlying atherosclerosis in acute and chronic HFpEF. Microcirculatory dysfunction is closely associated with atherosclerosis [27,28] and both are common in HFpEF [29,30,31]. We suggest that HFpEF hospitalisation and/or heterogeneity is associated with progressive atherosclerosis, which is strongly supported by the findings of the MESA trial [32], where visceral adiposity (the risk factor of atherosclerosis) in atherosclerotic patients without underlying cardiovascular diseases was found independently associated with hospitalisation for HFpEF during an 11-year follow-up period.

Intraplatelet Ca^2+^ elevation is important in the interplay between systemic pathogenesis and local (myocardial) abnormality in HFpEF. The progression of HF is related to increasing angiotensin-II formation in the heart as a result of enhanced LV end-systolic wall stress [33], a key determinant of LV diastolic function. The elevated intracellular platelet Ca^2+^ is well-characterised as a result of elevated angiotensin-II [34,35]. Therefore, our study likely confirms the connection between elevated intraplatelet Ca^2+^ and LV diastolic dysfunction in HFpEF for the first time.

Another key pathway is the regulation of IGF transport and uptake through IGFBP, centred by IGFBP complex acid labile subunit (IGFALS). The levels of IGFALS were significantly decreased in acute HFpEF vs. HCM. Given the pro-angiogenic function of the IGF pathway [36], our results highlight impaired angiogenesis as one of the potential pathogenic mechanisms across all phenotypes of HFpEF. This finding is supported by growing evidence suggesting impaired angiogenesis as the predominant pathogenesis of HFpEF [37]. The importance of angiogenesis is also demonstrated by perturbed LRG1 levels which follow a gradual increase from HCM to chronic HFpEF to acute HFpEF. The overexpression of LRG1 in HFpEF compared to healthy controls was demonstrated in another study [38], whereas our study is the first to report the differential abundance of LRG1 between different subtypes of HFpEF. Our findings suggest a key role for altered angiogenesis in HFpEF progression.

In terms of pathways perturbed in both acute and chronic HFpEF, we have identified amyloid fibre formation regulated by SAA1, which has also been detected in other multiomics studies using HF samples [8,19,20]. SAA1 is a major acute-phase protein mediating inflammatory amyloidosis. Comprehensive research studies recently discovered the presence of intramyocardial amyloidosis in HFpEF, which is characterised by deposition of wild-type transthyretin localised in the left ventricle [39,40]. Given that most patients with HFpEF often present with milder symptoms than HFrEF, the increase in SAA1 in chronic HFpEF group (compared to HCM) could be important in the clinical settings to identify patients with progression of HFpEF. During the acute-phase response, SAA1 acts as an exchangeable APO-like protein replacing APOs in HDL [41], which is consistent with the increased SAA1 coupled with decreased APOA1 and APOA4 in our study. Serum amyloid P component (SAP) is also reported being upregulated in acute and chronic HFpEF, stabilising the insoluble amyloid deposits by protecting fibrils from proteolytic degradation [42].

The vast majority of multiomics studies in the field of HF are mostly focused on HFrEF rather than HFpEF. A study by Raphael et al. [8] followed a similar experimental design to our study utilising human HFpEF plasma; however, the results were analysed in a non-quantitative manner. The only quantified protein identified in that study was S100A8. In our study, S100A8 was increased in abundance in acute HFpEF compared to HCM, strengthening the potential of S100A8 as a diagnostic target for HFpEF progression or hospitalisation in HCM patients.

Another HFpEF proteomic study utilised human myocardial tissues [7], specifically the homogenous HFpEF with cardiac-specific cause, similar to the HCM group in our study. The advantage of our study is that the plasma protein markers are more easily accessible and suitable for routine laboratory testing. Given that HFpEF commonly presents with substantial heterogeneity clinically, the inclusion of different types of HFpEF in our study could provide more informative insights into HFpEF pathogeneses. There are three significantly perturbed proteins shared between Chen et al.’s study [7] and ours, including SAA1, APOA4 and SERPINA3. Chen et al. [7] reported downregulated SAA1 and SERPINA3 in HCM compared to healthy controls; however, our data suggested increased abundance of these two proteins in heterogenous HFpEF compared to HCM. While in the Chen et al. study, APOA4 was upregulated in HCM compared to healthy controls, our study showed decreased abundance of this protein in a heterogeneous HFpEF group compared to a homogenous form of HFpEF or HCM. Although different comparison groups were used between this study and ours, it poses a question of whether certain biomarkers are differentially abundant between the local (myocardial) and circulating samples in HFpEF. Indeed, Li et al. showed that SAA1 was significantly decreased in myocardial samples collected from HFrEF patients [20], while SAA1 as an acute-phase protein is expected to be upregulated as shown in our study using plasma samples. Given that perturbation in the circulation is potentially an upstream pathogenic mechanism of HFpEF [23], understanding the differences in certain protein’s abundance between myocardial and blood-derived samples is important in evaluating the reliability of certain systemic biomarkers in reflecting pathogenesis of HFpEF.

Although this study provides new information for personalised monitoring of HFpEF patients, there are a few limitations. We chose not to employ any depletion methods to remove highly abundant plasma proteins to prevent co-depleting any differentially abundant proteins, an approach that is becoming the standard in plasma/serum analysis [43]. Follow-up studies will employ the more increasingly employed data-independent acquisition (DIA) with extensively fractionated pooled samples as a reference library to increase the depth of proteome analysis while managing instrument time. Validation cohort was not available for confirming differences in highly abundant proteins between different HFpEF phenotypes, which should be carried out in the future. In our study, we determined specific biomarkers that should be explored in larger studies in the future for better monitoring of HFpEF progression.

## 5. Conclusions

Our study is the first to apply comprehensive and quantitative proteomic screening of plasma samples from patients with different types of HFpEF, including HCM and acute and chronic HFpEF, representative of disease heterogeneity and hospitalisation. New potential protein markers were also identified for different HFpEF forms, including LRG1, SAA1 and ITIH3. We report perturbations in a series of systemic pathways including platelet activation and regulation of IGF transport and uptake. Many of these pathways are associated with microvascular inflammation, a well-established pathogenic process of HFpEF. We identified novel protein perturbations across HCM, acute and chronic HFpEF, including LRG1 and SAA1, which suggests the important roles of these proteins in acute exacerbation and hospitalisation of HFpEF. The network analyses showed the importance of ECM proteins in the pathogenesis of both heterogeneous forms of HFpEF (acute and chronic HFpEF) compared to the homogenous group with HCM. On the other hand, immune proteins appear to drive the acute decompensated HFpEF. In addition to biomarker potential, some of these proteins could have a therapeutic target potential, which should be explored in the future.

## Figures and Tables

**Figure 1 biomolecules-12-01419-f001:**
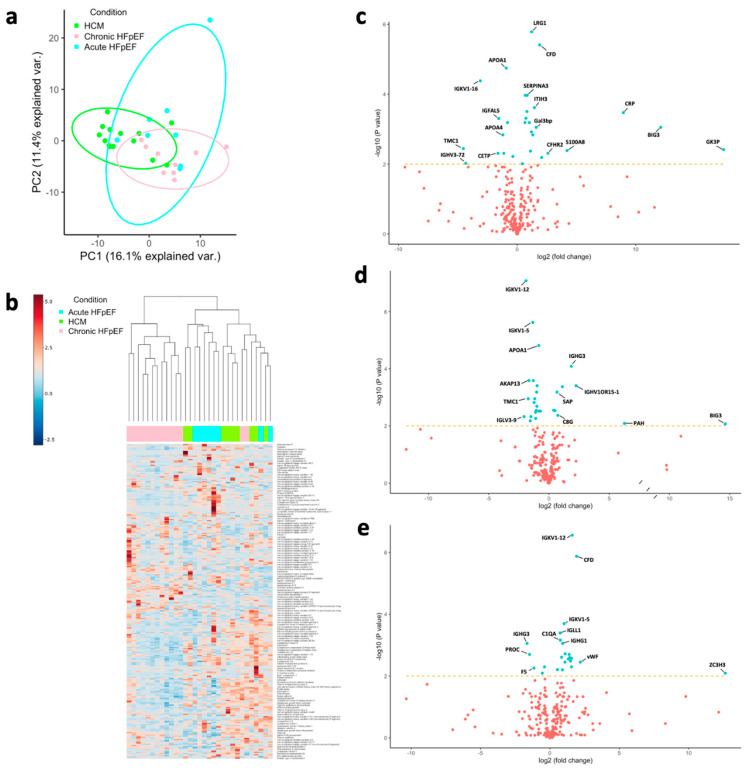
**Overviews of differential expression analysis.** (**a**) Principal component analysis (PCA) plot of proteomic data in HCM, chronic and HFpEF groups. (**b**) Heatmap of individual samples with hierarchical clustering dendrogram of proteome profiles across HCM, chronic and acute HFpEF groups. Volcano plots of proteomic data in (**c**) acute HFpEF vs. HCM, (**d**) chronic HFpEF vs. HCM, and (**e**) acute HFpEF vs. chronic HFpEF. Differential expression (DE) analysis was performed by fitting a linear regression model adjusted for age, gender and diabetes. DE proteins were defined as Benjamini–Hochberg adjusted *p* value < 0.01, as indicated by the proteins above the cut-off lines.

**Figure 2 biomolecules-12-01419-f002:**
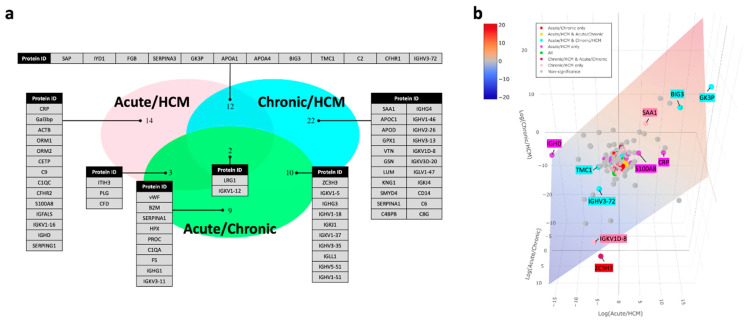
Highlighted proteins in differential expression analysis. (**a**) Triple Venn diagram summarising the differentially expressed proteins unique and overlapped between chronic HFpEF vs. HCM, acute HFpEF vs. HCM and acute vs. chronic HFpEF. (**b**) Three-dimensional plot of fold changes (Log_2_-transformed) of all identified proteins, with a regression plane filled with colour indicator of log_2_-fold changes.

**Figure 3 biomolecules-12-01419-f003:**
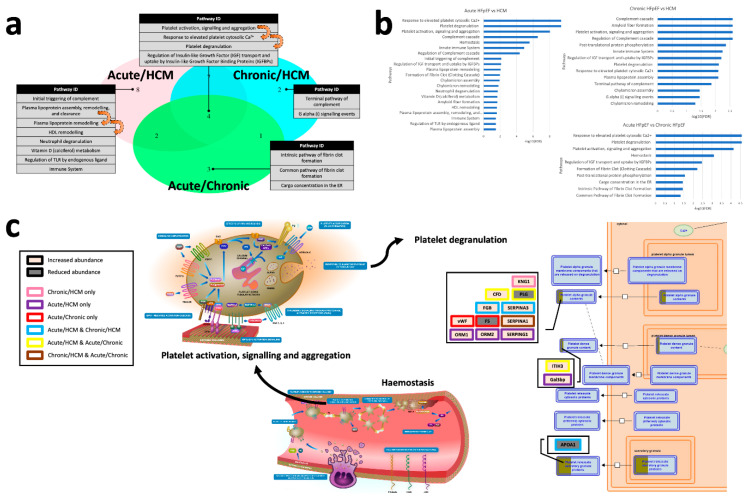
Pathway analysis. (**a**) Triple Venn diagram illustrating the number of pathways significantly changed in acute HFpEF vs. HCM, chronic HFpEF vs. HCM and acute HFpEF vs. chronic HFpEF. (**b**) Representation of all significantly altered pathways between the three comparison groups in order of significance (x-axis is −log_10_(FDR)); and (**c**) for the pathway of platelet degranulation in response to elevated platelet cytosolic Ca^2+^. IGF, insulin-like growth factor; and IGFBP, insulin-like growth factor binding protein.

**Figure 4 biomolecules-12-01419-f004:**
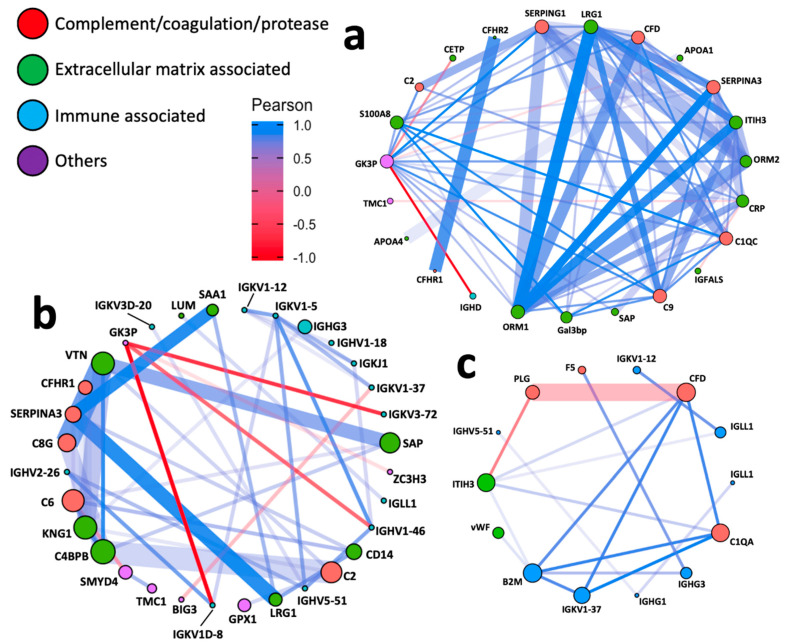
Network analysis. Network plots illustrating protein–protein interactions in (**a**) acute HFpEF vs. HCM, (**b**) chronic HFpEF vs. HCM and (**c**) acute vs. chronic HFpEF. Each node represents one protein; and each edge represents a pairwise Pearson correlation between proteins. The networks only show Pearson correlation coefficient > 0.7 or <−0.7. Node: colour coded according to the functional classification of proteins annotated by DAVID; size proportional to the corresponding eigenvector centrality (network influence). Edge: colour and length proportional to the Pearson correlation coefficient; width proportional to the scores derived from STRING database.

**Table 1 biomolecules-12-01419-t001:** Summary of patient characteristics.

Characteristics	Acute HFpEF (*n* = 8)	Chronic HFpEF (*n* = 9)	HCM (*n* = 14)
Age (years)	73.1 ± 14.2 ^¶^	64.6 ± 10.6	51.2 ± 14.0 ^¶^
Female (no. [%])	3 (37.5)	3 (33.3)	3 (21.4)
BMI (kg/m^2^)	31.4 ± 4.8	28.0 ± 2.5	26.1 ± 4.1
LVEF (%)	55.4 ± 10.2 ^¶^	57.4 ± 8.5	64.4 ± 4.1 ^¶^
NYHA class	II/III	I/II	I/II
Diabetes (no. %)	4 (50.0)	2 (22.2)	0 (0)
NT-proBNP (pg/mL)	15,417 ± 21,680 ^¶^	2266 ± 3032	3155 ± 3037 ^¶^
Echocardiography measurement			
LVEDD (mm)	56.5 ± 11.5	52.8 ± 6.9	47.8 ± 5.6
LVESD (mm)	38.3 ± 10.4 ^¶^	35.3 ± 8.3	29.1 ± 4.3 ^¶^
LAD (mm)	48.9 ± 5.9 ^¶&^	41.1 ± 3.6 ^&^	42.2 ± 6.6 ^¶^
Medications			
Statin (no. [%])	3 (37.5)	6 (66.7)	1 (7.1)
Beta-blocker (no. [%])	6 (75.0)	9 (100.0)	13 (92.9)
Calcium channel blocker (no. [%])	2 (25.0)	3 (33.3)	1 (7.1)
ACEi/ARB (no. [%])	5 (62.5)	7 (77.8)	4 (28.6)
Diuretic (no. [%])	4 (50.0)	4 (44.4)	4 (28.6)
Warfarin (no. [%])	1 (12.5)	1 (11.1)	0 (0)
Acetylsalicylic acid (no. [%])	4 (50.0)	7 (77.8)	1 (7.1)
Amiodarone (no. [%])	0 (0)	0 (0)	1 (7.1)
Isosorbide mononitrate (no. [%])	1 (12.5)	3 (33.3)	0 (0)

BMI, body mass index; HF, heart failure; HFpEF, heart failure with preserved ejection fraction; LAD, left atrial diameter; LVEDD, left ventricular end-diastolic dimension; LVEF, left ventricular ejection fraction; LVESD, left ventricular end-systolic dimension; NT-proBNP, N-terminal pro-B-type natriuretic peptide; and NYHA, New York Heart Association Functional Classification. ¶, *p* < 0.05 of a characteristic between acute HFpEF and HCM groups. *p* < 0.05 of a characteristic between chronic HFpEF and HCM groups. &, *p* < 0.05 of a characteristic between acute HFpEF and chronic HFpEF groups; statistical differences for each therapy and diabetes status were not calculated due a to small number of patients on each therapy or no patients with diabetes in HCM, respectively.

## Data Availability

The data presented in this study are available on request from the corresponding author and within the Appendix A provided.

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
