# Peer review of "Systemic Biomarkers and Unique Pathways in Different Phenotypes of Heart Failure with Preserved Ejection Fraction"

_biomolecules, 2022, doi:10.3390/biom12101419_

Round 1

Reviewer 1 Report

I read with interest the paper by Chen et al. After carefull reading I have only minor comments/questions for the authors:

1. Why the authos did not use an age- and gender-matched control group in order to show the differences between subjects without disease and patients with heart failure?

2. Do the authors believe that their findings on differencial protein expression in plasma samples have the potential to be used for screening purposes in the general population?

3. What is the power of the study to detect differences in plasma proteins? In other words is there any other possibility that more differences could be detected in a larger sample of patients? I suggest to mention the power of the study in the statistical methods section.

4. Table 1: I think that Stain should be corrected with Statin.

5. Is there any differences between male and females patients? 

6. Did the authors adjusted their results including medical therapy as described in Table 1?

Author Response

We sincerely thank the Editor and Reviewers for their time and consideration of our manuscript. We have carefully addressed all of their comments and modified the manuscript as described in the point-by-point responses below. This has improved our manuscript and it is now ready for publication.

Reviewer One:

I read with interest the paper by Chen et al. After carefull reading I have only minor comments/questions for the authors:

  1. Why the authos did not use an age- and gender-matched control group in order to show the differences between subjects without disease and patients with heart failure?

Response: We appreciate the Reviewer’s comment. However, the aim of this study was to provide better understanding of different forms of HFpEF including acute HFpEF given its heterogenous nature. As we stated on page 2, “the aim of this study was to identify perturbed proteins that could be translated into specific biomarkers for cases of HFpEF progression. We also identified associated pathways providing insight into the pathogenesis of different forms of HFpEF including HCF, acute and stable HFpEF patients.”

  1. Do the authors believe that their findings on differencial protein expression in plasma samples have the potential to be used for screening purposes in the general population?

Response: Yes, our manuscript will inform the future development of risk stratification biomarkers for acute and chronic HFpEF including HCM. We identified a number of promising new and known biomarkers that are biologically plausible, such as leucine-rich alpha-2-glycoprotein 1 (LRG1), serum amyloid A1 (SAA1) and inter-alpha-trypsin inhibitor heavy chain 3 (ITIH3). We have summarised in the Conclusion the most applicable findings with the potential for screening purposes and identifying those at risk of hospitalisation in HFpEF patients. Some of these biomarkers could have a therapeutic target potential as well. We have added a sentence now to this effect in the Conclusion section (page 11).

  1. What is the power of the study to detect differences in plasma proteins? In other words is there any other possibility that more differences could be detected in a larger sample of patients? I suggest to mention the power of the study in the statistical methods section.

Response: It is understood by the proteomics community that at least n=5 per group is needed to provide sufficient statistical power for most of the analyzed proteins if relative changes in abundances >1.5-fold are expected. The reference for this is now included as [18] - https://doi.org/10.3390/proteomes9040047. We have now included this information in the statistical methods section.

  1. Table 1: I think that Stain should be corrected with Statin.

Response: We apologies for this typo, it has been corrected now.

  1. Is there any differences between male and females patients? 

Response: The number of samples were too small to be able to look at the gender differences given that we only had ~30% females per group (Table 1).

  1. Did the authors adjusted their results including medical therapy as described in Table 1?

Response: We adjusted for age, gender and diabetes status, which are confounding factors for cardiovascular disease. We did not adjust for medical therapies due to a small number of patients on some therapies and we would need at least five individuals on the same therapy in a group to see whether therapy changes the biomarker levels. Furthermore, for this work to be translational and applied to general HFpEF population, given a wide range of therapies used in HFpEF patients, adjusting for medical therapies is not recommended.

Reviewer 2 Report

In this study, the authors used systematic and quantitative proteomic screening of plasma samples from patients with different subtypes of HFpEF. They also identified candidate biomarkers for acute and chronic HFpEF and HCM. Overall, this study is straightforward and informative. There is a minor point that hopefully can help improve the manuscript.

Figure 3C: It would be better to plot the pathway p-values for the most significant pathways as a horizontal bar chart.

Author Response

We sincerely thank the Editor and Reviewers for their time and consideration of our manuscript. We have carefully addressed all of their comments and modified the manuscript as described in the point-by-point responses below. This has improved our manuscript and it is now ready for publication.

Reviewer Two:

In this study, the authors used systematic and quantitative proteomic screening of plasma samples from patients with different subtypes of HFpEF. They also identified candidate biomarkers for acute and chronic HFpEF and HCM. Overall, this study is straightforward and informative. There is a minor point that hopefully can help improve the manuscript.

Figure 3C: It would be better to plot the pathway p-values for the most significant pathways as a horizontal bar chart.

Response:  We thank the Reviewer for their positive comments and the suggestion regarding Figure 3C. We have now included p-values of the most significant pathways as a horizontal bar chart and rectified Figure 3.